# Micromachined Thermal Time-of-Flight Flow Sensors and Their Applications

**DOI:** 10.3390/mi13101729

**Published:** 2022-10-13

**Authors:** Liji Huang

**Affiliations:** Siargo Ltd., 3100 De La Cruz Boulevard, Suite 210, Santa Clara, CA 95054, USA; liji@siargo.com; Tel.: +1-408-9690368

**Keywords:** MEMS flow sensor, fluidic property sensor, energy metering, microfluidics

## Abstract

Micromachined thermal flow sensors on the market are primarily manufactured with the calorimetric sensing principle. The success has been in limited industries such as automotive, medical, and gas process control. Applications in some emerging and abrupt applications are hindered due to technical challenges. This paper reviews the current progress with micromachined devices based on the less popular thermal time-of-flight sensing technology: its theory, design of the micromachining process, control schemes, and applications. Thermal time-of-flight sensing could effectively solve some key technical hurdles that the calorimetric sensing approach has. It also offers fluidic property-independent data acquisition, multiparameter measurement, and the possibility for self-calibration. This technology may have a significant perspective on future development.

## 1. Introduction

Micromachined flow sensors are one of the most successful devices in the MEMS industry. The first academic demonstration was presented by Putten and Middelhök in 1974. The sensor was built on a silicon substrate utilizing the anemometric sensing principle [1]. However, the commercial realization of the micromachined flow sensor only took place more than ten years later by Honeywell with the calorimetric sensing approach [2,3]. In the past half a century, many research reports have been published on micromachined flow sensors. Summaries of these activities can be found in several review articles from different periods [4,5,6,7,8,9,10,11,12,13]. Different physical principles have been applied to fabricate the flow sensing devices in the laboratories, including differential pressure, Coriolis, optical, and magnetic sensing. The most common approaches are thermal mass flow sensing. There are no moving parts in the micromachined thermal mass flow sensors, the structure is relatively simple, and the process is readily achievable with the MEMS process capabilities in commercial foundries and laboratories alike.

In the past 40 years, commercialization of the MEMS flow sensor has progressed steadily. More than ten companies are now offering their proprietary MEMS flow sensors for various practical applications in automotive, medical, utility, instrumentation, automation, and many others. However, MEMS flow sensors have not been attractive products for market analysts of the MEMS industry [14]. In many market reports of MEMS sensors, the value of the MEMS flow sensing products was either smaller than the actual size or was not considered as a market facilitator [15]. One of the reasons could be that commercial MEMS flow sensing applications are very fragmented. The total market value is not easily accounted for. Many MEMS flow sensors are not shipped in the form of a packaged die but inside fully functional end products in which the cost of the flow sensor die is trivial. The performance of the MEMS flow sensor involves complicated fluidic dynamics and the control scheme’s impact is also great. Therefore, the exact market value of the micromachined flow sensors cannot be easily evaluated. Nonetheless, the market for MEMS flow sensors has been growing significantly since its inception. Today, the annual shipment of micromachined flow sensor dies is estimated to be over 25 million, and the products are dominated by the thermal mass flow sensing principle [16]. Only a tiny percentage of the shipment of the products is based on Coriolis sensing principles [17,18,19,20,21,22,23,24]. The thermal mass flow sensors are manufactured with the calorimetric sensing principle, and anemometric sensors are relatively few. The micromachined Coriolis products apply only for microfluidics, and the shipment is currently limited due to the high cost. Some of the thermal mass flow sensor dies have been packaged for differential pressure measurements, and those are particularly successful in HVAC applications as the same pressure range offered by traditional pressure sensing products is of substantially higher costs. 

There are three high-volume applications for the current MEMS thermal mass flow sensing products on the market. The first successful one was for automotive mass air flow sensors (MAF) applied for fuel-efficiency control starting in 1995 [17]. In early 2000, applications in medical continuous positive airway pressure (CPAP) equipment [24] and utility city natural gas meters [25] also gained momentum and have been shipped in relatively high volumes since then. While MAF and CPAP applications do not require very high accuracy, and their working environments are relatively favorable, applications for utility natural gases have posed many challenges for the MEMS flow sensors. Utility gas meters are used for tariff purposes by city gas distributors, for which both precision and reliability are critical. Further, natural gases are mixed with different gases without a fixed composition. The gas composition variations will have a direct impact on the metrology precision of the calorimetric-based MEMS flow sensors. Another popular application is for medical equipment, particularly drug infusion-related equipment. It has been believed to be a potentially high-volume application for MEMS flow sensors, but the deployment is yet to be realized [12]. Similarly, the ever-discussed applications in microfluidics are also challenged by the fragmented requirements and problems in fluidic composition issues. Calibration with actual fluids for many microfluidic applications is not feasible. Moreover, the current trend for flow measurement, where ever the flow sensor is applied, is to improve the precision while acquiring as much information as possible for better control of a process. These demands ask for innovations in micromachined flow sensing technology. In this paper, thermal time-of-flight sensing technology will be reviewed and discussed. This technology is relatively less popular in both research and commercialization. However, it offers the capability to address gas composition variations and self-calibration that the current calorimetric or anemometric technology is unable to offer. With more commercialized products becoming available, it could be the preferred one to overcome some of the current key market hurdles for the abrupt growth of the MEMS thermal flow sensing applications. This paper will review the thermal time-of-flight technology and the micromachined sensor process and operation. The summarized historical research work will show and address why micromachined sensors can be the enabler of this technology. This paper will also discuss the potential volume applications of this technology. 

## 2. Thermal Time-of-Flight Sensing

Figure 1 shows the sketch of the thermal time-of-flight sensors. The classic concept (a) has a heater wire and a sensor wire. It is also named a pulsed anemometer, while (b), a micromachined sensor, often has a microheater and multiple sensors that are made on a thermally isolated membrane on the silicon substrate. The thermal time-of-flight sensing concept can be traced back to the late 1940s by Kovasznay [26]. A stationary hotwire was placed normal to the mean flow upstream of a movable sensing wire to measure the airflow in a wind tunnel. With the known hotwire driving frequency and comparing the sensing wire signal to the same, the spatial wavelength of the heat convection pattern was measured, and the air mean convection velocity was calculated. In recent years, quite a few excellent research works based on this concept have been published [27,28,29,30,31,32,33,34,35]. In these studies, the approach was also named pulsed anemometers. It generally measures the heat transfer transient time as well as the responses at each sensing element with the hotwire being driven with various heat waveforms. Several sensing elements can be placed downstream of the “pulsed hotwire” or the heater. Although these studies have detailed measurement-associated issues such as diffusivity, pulsating, laminar, and turbulent flow, most works discussed high-speed turbulent flow applications. Compared to calorimetry and anemometry, the advantage of thermal time-of-flight is to measure additional parameters besides the flow rates. As the thermal response of each sensing element is dependent on the fluidic thermal properties, thermal conductivity and thermal diffusivity could be obtained via the simultaneously measured thermal response data. With these data, fluidic dependency could also be possibly removed. Further, the transient time domain data are much more immutable to the background interferences. Despite the advantages, there are also apparent drawbacks. For the packaged hotwire sensors into a flowmeter, the data acquired from these sensing elements are still dependent on the type of fluid that flows through the enclosed conduit because of the thermal response of the wire. This fluidic dependence makes it no different from the other thermal sensing principles. The hotwire operating with the thermal time-of-flight mode will not be able to be packaged into a shielded tube as the anemometers for protection because of its requirement for time resolution. Vapors, particles, and other fluidic conditions are always a challenge for the reliability of those free-standing wires in the flowing fluid. The traditional hotwire design is also complicated for the alignment of the wires at the installation. Vibration and rotation (yaw) add errors during operation. Another disadvantage is that the hotwire thermal time-of-flight sensors have a smaller dynamic range as the diffusivity dominates the heat transfer at the lower flow speed. At the high-speed regime, the precise measurement of the time domain data was difficult due to the stringent requirement for a precision time resolution, particularly in the earlier studies when the electronics were not advanced enough. For the hotwire time-of-flight, the sensor also behaved quite differently at laminar and turbulence, making the control algorithm very complicated. Moreover, the mass of the traditional hotwire lacks the speed of response necessary for applications such as medical devices that require high-speed flow rate measurements. These problems would then create barriers to the commercialization of this technology. Products utilizing thermal time-of-flight sensing are only very few. Medical respiratory flow sensors for ventilators and wind tunnel laboratory measurement devices [36,37] are two known applications on the market. The medical flow sensor in respiratory applications is no longer a choice for new devices. Attempt for a natural gas meter using the hotwire thermal time-of-flight mode was not successful, likely due to reliability and high-power consumption, even though excellent dynamic range and gas independence were reported [38].

The heat wave propagation in the thermal time-of-flight sensing configuration observes the same physics for a thermal line source in a fluid. The total heat wave transfer includes both thermal diffusion and forced thermal convection, and the working principle can be expressed for energy conservation as below [29,33,39,40]: (1)$$\frac{\partial T}{\partial t}=\frac{k}{\rho c}\nabla^{2}T+\frac{Q\left(t\right)}{\rho c}-V\nabla T$$
where *T* (K) is the fluidic temperature, *t* (s) is the time, *k* (W m^−1^ K^−1^) is thermal conductivity, *c* (J kg^−1^ K^−1^) is thermal capacitance, and *ρ* (kg m^−3^) is the fluidic density. The heat wave *Q* (J) is a time-dependent value of either a modulated heat wave or a defined heat pulse. Therefore, the fluidic flow speed *V* (m s^−1^) can be obtained by solving the above equation. 

Apparently, for a static fluid, the flow speed will be null, and Equation (1) becomes
(2)$$\frac{\partial T}{\partial t}=\frac{k}{\rho c}\nabla^{2}T+\frac{Q\left(t\right)}{\rho c}$$

Some assumptions need to be applied to solve the above equations analytically. For a pulsed heat at the zero flow, the thermal diffusivity *α* of the fluid will be [39]:(3)$$\alpha =k/\rho c=V_{0}\;\left(x_{0}/4\right)$$

Alternatively, if a modulated heat H(t)∝1+cos(ωt) is applied [40]:(4)$$\alpha =k\;/\rho c=V_{0}^{2}\;/2\omega$$

In either of these heat modulation approaches, once the velocity of the constant phase (*V*_0_) is acquired at the static condition, the thermal diffusivity of the fluid can be obtained. The *x*_0_ could be considered for the characteristic length of a known fluid within which the diffusion would be dominated, or it defines a low flow rate detection limitation. This is an obvious advantage for the thermal time-of-flight approach since the physical property of the fluid can be simultaneously measured. The calculated and measurement data agreed quite well per the studies [41]. On the other hand, it could also be used to optimize the design for the distances between the heat source and sensing elements for a specific application.

According to the numerical and analytical study on the thermal time-of-flight sensor [34], the time shift (time-of-flight) related to the flow velocity depends on both advection and convection shifts. The advection time shift is nearly proportional to the reciprocal of the flow velocity with some nonlinearity added by thermal diffusion. As the heat transfer coefficient is also proportional to the flow velocity, the convection time would be the reverse proportion to the flow velocity. Thermal convection time shift becomes insignificant when the flow velocity is very low. In addition, when the hot wire diameter decreases to several tens of micrometers, the effect of convection would also be negligible. Using a finite volume method, the study found that simulation data can match well to the analytical results of Equation (1) when the hot wire diameters are within a few tens of micrometers. Hence, at zero flow conditions, the measured time shift would be contributed by advection and kept as a constant or the fluid’s diffusivity can be measured accurately at this condition. These results also suggest that micromachined sensors would be more favorable compared to the traditional hot wires for a less complicated heat transfer since thermistors with micrometer dimensions are readily achievable with today’s micromachining process. 

A one-dimensional approach could be applied for the micromachined sensors as the contributions from other dimensions would be very limited. This approach would also make the analytical solution easier, and the result of such can be expressed as [39,41]:(5)$$T\left(x,t\right)=\left(\frac{Q}{4\pi kt}\right)\exp\left(-\frac{{\left(x-Vt\right)}^{2}}{4\alpha t}\right)$$
where *Q* is the heat source, such as a pulsed signal, and *k* is the thermal conductivity. *V* is the averaged flow velocity and *α* is the thermal diffusivity.

These analytical results also suggested that gas thermal properties can be extracted from the constants of the dynamic measurement data. On the other hand, if the sensor only has one modulated microheater and one sensing element for the measurement, the acquired data will still be dependent on the fluid thermal properties. Heat transfer will occur at the substrate. Diffusive spreading of the modulated heat signals, conduction via the boundary layer, and the intrinsic thermal responses of the sensing elements will all contribute to the measurement. Therefore, the value of a thermal time-of-flight approach would require additional sensing elements to remove fluid properties. Simple calibration with any accessible fluid, such as air or water, can be applied to measure other fluids without losing the metrology accuracy. Multiple sensing elements are readily achievable for the micromachined sensors, where the sensing element arrangement can be well defined with the photomasks that guarantee the desirable reproducibility. Assuming a micromachined sensor with two sensing elements, and the distances of these two sensing elements from the microheater are *x*_1_ and *x*_2_, respectively, solving Equation (5) can remove the fluid property-related parameters and obtain both the thermal diffusivity, *α*, and fluidic property independent flow velocity, *V*, as shown in Equations (6) and (7), respectively [39,41]:(6)$$\alpha =\frac{\left(t_{1}^{2}x_{2}^{2}-t_{2}^{2}x_{1}^{2}\right)}{4\left(t_{2}t_{1}^{2}-t_{1}t_{2}^{2}\right)}$$
(7)$$V=\sqrt{\left(\frac{x_{2}^{2}}{t_{2}}-\frac{x_{1}^{2}}{t_{1}}\right)/\left(t_{2}-t_{1}\right)\;}$$
where *t*_1_ and *t*_2_ are the heat transfer time recorded at the two sensing elements, respectively.

The differential temperatures between the sensing elements or amplitude data from each of the sensing elements can also be acquired simultaneously in addition to the time domain data. Therefore, the calorimetric or anemometric data can be measured at the same time. The calorimetric or anemometric data will provide a mass flow rate similar to the conventional calorimetric or anemometric approach per the data acquisition process. In the microfluidic flow measurement, the liquid is generally non-compressible, and the pressure effects of compressibility can be considered as secondary. Liquids have a much larger heat capacitance than gases, making the sensing element resistance-related temperature effects less pronounced. Moreover, most importantly, the dynamic measurement range can be substantially extended with the multiple sensing elements on a single chip. On the other hand, the pressure effects can be used to calculate the fluidic pressures for the gas flow measurement by correlating the time domain and amplitude data.

In the quasi-static situation, the cross-over flow velocity from the diffusion to time-of-flight could also be calculated in the one-dimensional approach, and Equation (3) could be used for the estimation. 

Another advantage of the micromachined sensors over the traditional hot wire is the much lower power consumption or a lower temperature elevation in the fluid. External power injection is particularly sensitive for microfluidic applications where quite some liquid would be temperature-sensitive. For the micromachined sensors with a modulated microheater, the constant heating spot in the flow channel is avoided compared to a calorimetric sensor where the microheater could create a heating spot. Micromachined sensors are often made on a thermally isolated membrane with precise sensing element spacings, and with multiple sensing elements, and also extending the dynamic range. Commercial MEMS calorimetric flow sensors with membranes have demonstrated excellent reliability in many applications. In summary, the thermal time-of-flight sensor can measure mass flow, fluidic velocity, and fluidic properties, making the technology ideal for solving many issues in applications that the current MEMS calorimetric or anemometric sensor could not furnish. 

## 3. Summary of Micromachined Thermal Time-of-Flight Sensors

It could be that the earlier research data for the pulsed hot wire anemometry and the more complicated control electronics, as well as the data processing scheme, make the thermal time-of-flight sensing technology less favorable for both commercialization and research alike. In the previous review articles for micromachined thermal flow sensors, thermal time-of-flight technology was only addressed briefly, or in some review articles, it is even completely excluded [4,5,6,7,8,9,10]. Table 1 summarizes the literature on micromachined thermal time-of-flight sensors since 1986. Although the studies are few compared to those for MEMS calorimetric and anemometric flow sensors, some critical issues of the technologies have been well addressed in these papers. The aspects and results of each work are subsequently discussed. 

The first micromachined thermal time-of-flight sensor on silicon could be the one reported in 1985 by Lambert and Harrington [40] that targeted the application of automotive air-fuel efficiency. The authors believed the time difference measurement would relax many system accuracy constraints. The sensor was made on a silicon substrate coated with polyimide for thermal isolation. Lead (Pb) metal film with a line width of 39.4 µm was driven by a 154 Hz sinusoidal source to create the temperature oscillation, and lead telluride (PbTe) thermocouples located at about 150 µm from the micro heater were used for the detector. The sensor was packaged into a 6 mm diameter flow channel, and airflow of 0~250 g/s (~12,630 L/min) was measured. The study is particularly valuable by its detailed data on the effects of thermal isolation as the heat transfer takes place in both the flow medium and the sensor substrate. A proper sensor design would minimize heat transfer effects in the substrate. The thermal diffusivity of the medium, in which the heat transfers, determines the exponent decay rate of the modulated heat with distance. The experimental data presented in the study matched well with the theoretical model that the authors presented. 

Stemme [42] fabricated a CMOS sensor on a silicon beam with polyimide as the support and thermal isolation. The chip was heated by a pulsed modulated square wave resulting in electrical dissipation in two bipolar transistors. A diode measured temperature responses on the same chip. In an experiment for air flow measurement, the modulating temperature was switched from 96 to 146 °C above the air temperature. Moreover, the heating and cooling responses were recorded. The data fitted well with the theoretical predictions according to the oscillation differential equation of the sensor chip temperature. By examining the results for the airflow in the range of 2 to 30 m/s, the linearity of the output was better than the sensor operated with the differential temperature mode for the same sensor [68].

Inspired by the success of a thermal time-of-flight sensor configurated with individual glass-encapsulated thermistors for microfluidics [69,70], Branebjerg et al. [43] and Yang et al. [44] fabricated a monolithic silicon flow sensor with a heating diode, a measurement diode, and a reference diode. The three diodes had the same size of 250 × 200 µm, and the distance between them was 1500 µm. The sensor was packaged into a liquid flow channel with a cross-section of 1500 µm square and operated at the thermal transit time mode with a frequency from 2 to 5 Hz. The sensor successfully detected 100 µL/min water flow with excellent stability and an estimated accuracy of 0.2% at the 200 µL/min. Compared to the individually encapsulated thermistors, the micromachined sensor had a significantly better dynamic performance by a factor of 90 (response time reduced to about 1 s from 90 s). However, the data also indicated that this sensor’s dynamic measurement range would be limited due to the weak signal at the low flow and the resolution issue at the high flow. 

While developing a microflow device for microfluidics, a thermal time-of-flight sensor was fabricated to provide feedback to control the microvalve and pumps in the system [45]. The sensor was made on a 0.2 mm silicon substrate composed of one microheater and three downstream sensing elements. These thermistors were made with platinum and had similar structures and electrical resistances. The thermistors were made on a diaphragm of the silicon oxide and silicon nitride membrane supported by a silicon frame heavily doped with boron for isolation and stress balance. The distances of these three thermistors from the microheater were 1.0, 4.2, and 10.2 mm, respectively. The data acquired from the three sensing thermistors could detect a gas flow speed of up to 5 m/s with singularity from the closest sensing element at about 4 m/s. There were about 10 times differences between the calculated and the measured flow speed by the sensor, but the authors did not offer an explanation. 

Compared to the traditional pulsed wire anemometer, a micromachined anemometer was believed to have a better performance because of its better thermal response. A micromachined hot-wire anemometer was designed with a similar structure to a conventional counterpart. [46] The sensing wire of the micromachined anemometer was made with heavily doped polysilicon, about 0.5 μm thick, 1 μm wide, and 10–160 μm long. In addition, a silicon beam and a thick Si handle were also made with the silicon wire. The Si beam was designed for a thermal and mechanical buffer between the supports and the handle to avoid interference with the flow. Like the traditional hot wire, the sensing wire was also made free standing to optimize the interaction with the flow and to minimize the thermal conduction to the handle of two parallel supports. Depending on the silicon wire length, a time constant of 5 to 50 µs, corresponding to a wire length of 10 to ~140 µm, could be achieved. 

To solve the problem that an anemometric or calorimetric flow sensor is fluidic composition dependent or is unable to measure a mixed gas (H_2_/N_2_), the thermal time-of-flight sensor was developed to be placed together in parallel to an anemometric flow sensor [47]. While an anemometric sensor outputs the signal related to the mixed gas composition, the thermal time-of-flight sensor would have additional time domain information. The authors indicated that the thermal time-of-flight sensor would not measure the gas-independent flow rate. However, they could apply an “artificial neural network” to obtain the desired results, although the detailed approach was not presented. To achieve the desired performance, the time-of-flight sensor made on silicon was a constituent of an array with as many as nine thermistors with a distance of 80µm between the thermistors. Each thermistor could be dynamically switched between the energy dissipating or anemometric and temperature-dependent resistive sensing function. Measurement of the different gas mixtures of helium and nitrogen showed a dynamic range within 10:1 could be achieved for a full scale of 20 mL/min in a flow channel of 250 × 1000 µm. In the paper, the authors also noticed that the thermal time-of-flight sensor was incapable of measuring of very low flow rate where the thermal diffusion dominated. With a similar sensor (spacing between the sensing elements was 100 µm with a length of 300 µm), the same research group further explored the thermal time-of-flight approach. The detailed theoretical explanations in this paper have been used for quite a few related research works. It revealed that the measurement principle could also be used to sense the thermal properties of the fluid, including thermal conductivity and thermal diffusivity [39]. 

Ashauer et al. [48] used a similar approach of dual sensors made of combined calorimetric and thermal time-of-flight elements to measure the microflow. However, the study is for the extension of the dynamic measurement range: a calorimetric sensor covered the low flow measurement while a thermal time-of-flight sensor was applied for high flow. A proper design of the two sensors and intelligent calibration must be applied to ensure a proper overlap of the two outputs. The micromachined sensor was made with polysilicon as the microheater, and thermocouples of polysilicon and gold were used as the sensing elements for both these two types of thermal sensors. The distance between the microheater and the thermocouples was much longer for the thermal time-of-flight sensor. These thermistors were placed on a very thin silicon nitride film for support and thermal isolation. It was estimated that the film was able to bear more than 1 bar pressure under stable conditions. With this approach, a dynamic range of 1500:1 was claimed. However, when applied to different fluids, the data showed the measurement was strongly dependent on fluidic properties at the low-flow calorimetric sensing regime and even for the data acquired at the lower flow rate from the thermal time-of-flight sensor. The dynamic range of the fluidic property-independent time-of-flight sensor would be limited to 3:1.

Instead of packaging separated calorimetric and thermal time-of-flight sensors into the same flow channel for additional measurement advantages, a sensor array was fabricated and operated at all thermal sensing modes of calorimetric, anemometric, and thermal time-of-flight sensing [51,71]. The sensor array was made with several equally spaced (500 µm) platinum wires on top of a parylene membrane, and the bottom side of the parylene was used as a cover to seal the microfluidic channel; hence, the sensor was considered to be “non-invasive” to the fluid under measurement. In the time-of-flight mode, a single pulse of 3 Vdc was applied to drive the micro heater (one of the thermistors in the array), and the data were taken from another thermistor 1 mm apart. The sensitivity of the sensing elements could be optimized for the pulse frequency. By comparison of the data for the three different thermal sensing modes, time-of-flight seems to require the highest power or incur a higher temperature elevation. At the same time, it would be less sensitive at the low flow. In addition, it had a smaller dynamic range with some nonlinearity in the data acquired. Another report [60] confirmed the higher power when operating with the time-of-flight mode. Compared to the calorimetric mode, the time-of-flight mode could measure a much (2×) higher flow speed. The calorimetric and anemometric data in all these studies showed consistent results for their well-known characteristics. Dual-mode operation with both calorimetric and thermal time-of-flight on the same micromachined sensor was also applied for power saving [56,72]. In the study, germanium was used as the sensing thermistor, and chromium was made for the microheater and placed in the middle of the two-sensing thermistor with an equal distance of 675 µm, all on a silicon dioxide and silicon nitride-combined membranes for thermal isolation. This structure allowed for low-power consumption even for calorimetric mode at a maximum of about 5.4 mW (4 V, 1.35 mA) in constant heating. With the time-of-flight mode, the duty cycle can be extended for a steady flow, and power consumption of about 2 mW could be achieved. However, the authors also noticed that there would be a low flow end cutoff using time-of-flight mode due to the domination of thermal diffusion. 

In an attempt to make the sensor more robust to contamination, a thick film thermal time-of-flight sensor was proposed [49]. The sensor was made on an alumina substrate, and glass was used for thermal isolation. Pt/Au was screen-printed as the microheater, and either nickel or pyroelectric sensors (PZT) were also screen-printed. Compared with the micromachined silicon sensors, the thick film sensor was rather large with a dimension of 25 × 15 × 1 mm^3^, and the sensor could have a size of 1.5 × 2 mm^2^ with the sensor to microheater distance of about 5 mm. Although the report did not present the contamination-related reliability data, it showed an achievable full-scale accuracy of 2.5%. The sensor also had a much slower response of within 5 s and a power consumption of more than 2 W. This high-power consumption was undoubtedly unfavorable for some applications where the elevated temperature could impact the fluidic properties. Another work [56] fabricated a silicon-based titanium/platinum sensing array for use with corrosive gas. The data showed that the thermal time-of-flight mode was comparably better for the rangeability with a millisecond response. Simulation on thermocouples with grounded stainless steel found that accurate measurement can be achieved with the time-of-flight mode and an auto-adaptive impulse response function [73]. Therefore, a proper package would also be possible for the micromachined sensors to apply for the measurement in harsh environments.

Two-dimensional numerical simulations using ANSYS/FLOTRAN were compared to the experimental data from a surface micromachined thermal time-of-flight sensor [52,74]. The sensing elements were made of phosphorus-doped polysilicon, and the distances between the microheater and the sensing elements varied from 210 to 1038 µm. The sacrificial layer was made with PECVD oxide of about 1.2 µm. For the simulation, the heater was placed into a 1 mm diameter flow channel with a heater temperature of 200 K and a heating pulse width of 200 ms. The heat convection and conduction model were adapted from a previous publication [75], where a quasi-static situation and temperature variation only confined to one dimension were assumed. It was likely that the heating or the elevated temperature in the tiny channel was too high. The best sensitivity was numerically and experimentally confirmed at the fastest distance in this study. In addition, the measurement data exhibited a constant positive deviation from the numerical data. It further had another positive deviation when compared to the theoretical calculations. Hariadi et al. [50] noticed that for a micromachined CMOS thermal time-of-flight sensor operated at the pulsed mode, the device geometry, power level, and pulse characteristics would all impact the final device performance. A sensor with a single heater on a thermally isolated membrane and a diode on the silicon substrate was used to measure the fluid temperature. An approach with a model of the thermal boundary layer was used to decompose the heat transfer in the fluid and sensing element substrate. Based on the composite mode, the simulation was carried out using SPICE and analog hardware description language (HDL). The results indicated that a smaller pulse width would have a better resolution and a larger dynamic range. As an infinite pulse width would equal the constant power mode, the results indicated that the thermal time-of-flight mode could extend the dynamic range by properly selecting the pulse width compared to the data presented for the same sensor on a constant power anemometric mode [76]. Another finding from this simulation was that the data were based on a monocrystalline silicon membrane of 5 µm thick with which the thermal time-of-flight mode could be well applied. Therefore, although the substrate heat transfer could impact the performance, a superior thermal isolation membrane (such as silicon nitride) would not be necessary for the device to be operated in thermal time-of-flight mode. 

For many microfluidic applications where flow measurement is required, the micromachined thermal flow sensors are the first choice if the application is cost sensitive. With a proper adjustment of the circuitry, the same sensor could be applied for both gas and liquid. A report used the same thermal time-of-flight sensor for gas flow for a city natural gas flow meter prototype and a manifold-packaged microfluidic product for medical applications [57]. The micro heater and sensing elements would be further thermally isolated for the gas flow sensing with open slots on both sides of these thermistors. However, the same could not be applied to liquid measurements. Therefore, if the sensor applies to both gas and liquid applications, there would be some thermal isolation sacrifice in the gas flow measurement. The reported data were acquired from a sensor with an asymmetrical design of the distance for the micro heater with respect to the up and downstream sensing elements of 200 and 250 µm, respectively. For this configuration, the outputs showed that the fitting was far different from the calculated theoretical values, although the measured data had good repeatability. This observation was likely due to the pronounced heat transfer via the membrane. The structure would also limit the applicable pressure to maintain an undeformed membrane flatness during the measurement. The data showed that the flow speed measured with the same sensor in the air could be 500 times higher than those in a microfluidic (water) medium, which was in agreement with the thermal property differences between air and water.

With the increasing research activities in microfluidics, applications of a micromachined thermal time-of-flight sensor were further explored. Berthet et al. [58,77] fabricated a glass/silicon/glass device using a bulk micromachining process for microfluidic application. In this study, the micromachined flow sensor was composed of a suspended heater and more than one sensing element across the microchannel with a dimension of 100 × 500 × 1000 µm. The structure had no membrane that reduced the unwanted heat transfer via the supporting membrane. The process was done with an SOI wafer, and the channel was formed via an anodic bonding of another glass wafer. While the data showed sequences of heat pulses and an estimated 5-degree elevation in temperature to ensure no impact on the fluidic properties, the heat dissipation from conduction via the channel wall would still take place, in particular for the signals acquired from sensing elements with a larger distance from the heater. However, with the differential measurement of two sensing elements at the different spaces from the heater, the imposed flow velocity can be linearly correlated to the measured velocity. The sensor also achieved a dynamic range of 1400:1. The authors proposed a phenomenological model and did computational fluid dynamics (CFD) simulations with the commercial software Ansys (FLOTRAN) and COMSOL. The analytical result for the temperature evolution in one dimension was slightly different from the previous results [39,41]. Although the fluid-independent analytical results were not explicitly presented, the experiment data did show that the differential measurements from two sensing elements could yield a similar correlation (6% deviations) between the imposed flow velocity and measured ones for water and hexadecane. These two liquids had a difference in diffusivity of 35%. 

Since the thermal time-of-flight approach directly measures the time domain data of heat transfer and the distance between the heat source and the sensing element, which can be well-defined on a micromachined sensing chip, there would be a possibility that the flow speed can be calculated directly from the acquired data without the need for calibration, which is not possible for the other thermal flow sensing approaches. In practice, there will be other difficult factors that will affect the direct calculation or the measurement of a pure flow speed-related time difference. There is still hope that some correlation could be revealed once the other factors become a fixed constant or are measured beforehand. In a series of studies based on hotwire time-of-flight [78,79] for a calibration-free or self-calibration of a thermal time-of-flight sensor and comparison of experimental data on both air and water, it was found that the “calibration-free” thermal time-of-flight measurement principle needed to be combined with conventional anemometry. Further, the applicable rangeability was limited. Another claim of a micromachined sensor [53,80] with two conductive loops suggested that direct measurement of the electrical outputs of the pair could achieve a calibration-free flow speed measurement. However, the exact results were unknown, and any such products have not been seen on the market.

Cross-contamination is a big concern for many medical or food and beverage applications, and disposable products are greatly appreciated. With the potential of self-calibration and multiparameter detection capability, a cost-effective micromachining process was studied with the thermal time-of-flight sensing technology for microfluidic applications. The thermal time-of-flight sensor was fabricated using a screen-printing approach, in which the sensor had a silver micro heater and a downstream thermocouple that was composed of carbon black and silver particles. These sensing elements were made in a polymer mix on the glass substrate with a 3 mm space between the micro heater and the thermocouple [63]. The heater was a 100 µm width meander. The sensor substrate was also used as the base of the microfluidic channel, which simplified the assembly and made the device very cost-effective. A drawback of this approach would be the prolonged time delay compared to the chips made on silicon with a much narrower line width for the micro heater. The slow time response could be an issue for some applications. Another low-cost fabrication approach [66] of a microfluidic thermal time-of-flight sensor was made on multilayer low-temperature cofired ceramics with gold/platinum composite for microheater and negative temperature coefficient (NTC) materials for the other four thermistors. One thermistor was placed upstream to combine one of the other three downstream thermistors for calorimetric sensing for low flow rate detection, while the three thermistors were for a time-of-flight measurement. The data showed a much faster response than those in [81], likely due to a better thermistor design, and the results fitted well with an empirical equation. However, the time delay was still longer than the sensors made on a thermally isolated substrate. 

To reduce the power consumption and further improve the signal-to-noise ratio such that a better accuracy for the thermal time-of-flight sensing technology, sensors with heat emitting nano filament (100 × 1000 nm^2^ in cross-section) and sensing nanowires with small pn-junctions (800 × 100 nm in cross-section) were proposed. The nano-heater to nanowire space was 6.5 µm, and the space between the nanowires of 1.9 µm was fabricated on the silicon substrate. Simulation indicated that up to 2 m/s nitrogen flow could be assessed, but no experimental test data were presented in the literature [61,82]. 

The capability to simultaneously measure fluidic flowrate and fluidic properties in thermal time-of-flight sensing technology has encouraged more research efforts. As discussed earlier, the results presented earlier [39,47] had not been convincing enough for a practical realization. Studies were also carried out for the static fluidic conditions to measure fluidic thermal properties alone. A micromachined calorimetric sensor was placed into various common liquids, the chromium microheater was driven by a sinusoidal heat wave, and data were collected for the amplitude of excess temperature (thermal conductivity) and phase shift (thermal diffusivity) against the frequency from 1 to 10 Hz. A 2-dimensional analytic model was used to assess the data acquired with a reasonable agreement between the measured and analytical ones [54]. Using a similar sensor, the studies were extended for nitrogen gas, which also showed reasonably good agreement for the measured data and analytical model. For the gas measurement, a higher frequency scan was applied from 10 to 2 kHz, and data showed a better correlation below 400 Hz [55]. The same approach was applied to measure the concentration of carbon dioxide and nitrogen mixed gas via the measured thermal properties [59]. The measured phase shift had an excellent linear correlation to the carbon dioxide by up to 10% in volume percentage. The data also indicated that the measured diffusivity had a strong dependence on the micro heater driving frequency (results presented for a driving frequency from 70 to 140 Hz). This made the measurement procedure complicated as the measurement for different gases (nitrogen or carbon dioxide) would require a different driving frequency for better accuracy. Moreover, such a frequency could not be predetermined. In another study, a uniquely designed and micromachined sensor was composed of a germanium thermistor surrounded by four arc-like central heater elements made of chromium. The other four germanium thermistors were symmetrically located around the micro heaters. These thermistors were fabricated on a 1.4 µm thick, 1 mm wide silicon nitride/oxide diaphragm [62,83]. This structure enabled the authors to reveal in detail the heat transfer competition between diffusion and forced convection, both experimentally and theoretically. The data showed that the thermal conductivity of the fluid could be extracted via the velocity-independent temperature phase, which could be further applied for correction in the temperature amplitude for a “medium-independent” flow measurement. As such, it could serve similarly to the approach in [59] to determine the concentration of a binary gas mixture. Some recent works [64,84] employed micromachined thin wires (2000 × 6 × 0.3 µm) suspended on a silicon trench of 2000 µm width and 300 µm deep. The micro heater wire was made of AuCr, and the sensing wire was made of Ni. This structure allowed the sensor to have a very fast response of 0.5 ms and relatively low power consumption. With the sensor operating at the pulse mode, thermal conductivities of methane-hydrogen mixture with several different concentrations correlated well with the thermal responses of the sensor, although the data could only be acquired at no-flow conditions. The measured velocity was still gas composition dependent, but the square root of the velocity was linearly correlated to the temperature. This observation was likely due to the sensor’s design, and the data at very low flow velocity was unavailable. 

## 4. Micromachined Thermal Time-of-Flight Device Design, Fabrication, and Operation

### 4.1. Device Design Considerations

As summarized above, thermal time-of-flight sensing can be realized via quite different designs. There seemed not to be a preferred one among all the approaches reported. Ideally, the time-of-flight concept asks for an anemometer design in which the heater and sensor will have a small mass of negligible thermal responses to the medium and reach a true time-of-flight sensing independent of fluid compositions. Practically, such a design is not feasible. With micromachining approaches, the design and realization of the thermal time-of-flight device become closer to the ideal concept. Hence, it could be the ideal enabler of this technology. The unwanted effects of vibration, alignment, and yaw (rotational effects) in a traditional design could all be eliminated with the micromachined approaches. The thermal response of the elements and the supply power can also be significantly reduced, and the reliability can be much improved. These features significantly promote the feasibility of commercial products. 

Because of the diversity of the applications, the design of the micromachined thermal time-of-flight sensor will be application dependent. Key parameters include the microheater and sensing element linewidth, the numbers of the sensing elements, and distances between the microheater and sensing elements. The linewidth will determine the thermal response. A narrower linewidth will help to have a faster response and less fluidic property-related response, but it will reduce the signal-to-noise ratio. It will also be limited by the input power of the specific applications. To achieve multi-parameter detection and take the advantage of the thermal time-of-flight measurement principle, the sensing elements downstream of the microheater should be at least two. The distances between the microheater and sensing elements will also be a consideration for the specific applications. A smaller distance will have less “time-of-flight” signals as thermal diffusivity will play a major role. A bigger distance will have stronger time-of-flight signals, but it will also require a higher heater power and will have a smaller dynamic range as the signals will quickly decay to lose the resolution for a reasonable data acquisition. The liquid applications will require much higher power than those for gases. However, for microfluidic applications, a high power would sometimes be detrimental as it could alter the fluidic properties. Therefore, the design needs to have a comprehensive consideration for a specific application. Thermal isolation should be another key parameter for the sensor design. For a micromachined sensor, a membrane is often used for thermal isolation. Open slots on the membrane near the microheater and sensing elements should be designed for gas flow applications. For liquid flow, alternative thermal isolation materials must be taken into account. 

The sensing elements can be designed using one of the three common thermal sensing approaches, i.e., thermoresistive, thermoelectric, and thermoelectronic sensing. The schematic structures of these devices are shown in Figure 2. Detailed discussions of these three sensing mechanisms can be found in the literature [5,7]. Thermoelectronic sensing design utilizes semiconductor junction diodes as the sensing elements, e.g., simple bipolar junction transistors. The process is CMOS compatible and easy to be fabricated in a miniaturized format. The temperature sensing mechanism is well understood, and its sensitivity could be easily tailored. However, its thermal isolation process would be relatively complicated, with the limitation of the thermal process control efficiency undefined. Furthermore, the subsequent calibration required great attention. These sensors mostly appeared in the earlier literature and are not the choice for commercial products. 

Thermoelectric sensing, on the other hand, has several advantages. Its temperature sensing capability is offered by microfabricated thermopiles or multiple connected thermocouples with which a voltage will be generated when a temperature difference exists across the two ends of two connected dissimilar electrical conductors. Both connected ends of these conductors form an electrical junction. As the thermocouple can be made with doped polysilicon or CMOS-compatible metals, the thermoelectric sensors can also be fabricated via the CMOS-compatible process. With the state-of-the-art semiconductor process, a thermocouple’s size can be made much smaller. Therefore, in a fixed area, the number of thermocouples can be increased, or the sensitivity of the thermopiles can be significantly boosted. Because the thermopile is a thermal energy harvester, with proper thermal isolation and optimization, the common drifting problems associated with thermal sensing could be significantly reduced. Therefore, the resulting sensor can be very much desired for practical applications, particularly low-power applications. This sensing approach has been adopted by the design of many commercial MEMS calorimetric sensors. However, the design with multiple thermopiles would be more complicated for thermal time-of-flight sensing as thermopiles require both hot and cold junctions that limit the spacing design of any two sensing elements. 

Thermoresistive sensors have merit for their simplicity in fabrication, and a broad spectrum of material selection is available for today’s commercial MEMS foundries. For a thermoresistive sensor, heat transfer or temperature variation will cause the resistance change of an electrical resistor because of its intrinsic temperature coefficient. Semiconductor materials such as doped polysilicon can also be used; hence, such sensors can be made with the CMOS-compatible process [85]. The thermoresistive sensor can have a high sensitivity and good signal-to-noise ratio with the proper selection of materials. It is the technology for earlier commercial thermal calorimetric or anemometric sensing products. The structure of a thermoresistive sensing element allows it to be easily duplicated. Hence, this technology would be preferred for the thermal time-of-flight technology to realize its multiparameter capabilities with the multiple sensing elements. One of the disadvantages of the thermal time-of-flight sensor is that it is unable to acquire flow speed data for very low flow speeds where thermal diffusion dominates. For the desired sensor with a large dynamic range, a pair of thermistors can be placed close to the microheater and operated in the calorimetric mode for metering the low flow speed. 

The spaces between the microheater and the sensing elements depend on a specific application with full-scale flow speed and power consumption consideration. The selection of the materials for the microheater and sensing elements would be more for the reliability and sensitivity requirements. For general-purpose applications, for example, the calorimetric sensing elements to the microheater should be anywhere from 5 to 60 µm, which will allow a measurement of approximately to cover 0.01 to 30 m/s for air. The final results will depend on the signal conditioning and control electronics as well as the algorithm for the data processing. Therefore, in the combined sensing application, the calorimetric sensing elements should be placed closer to the micro heater to have the highest sensitivity for low flow speed where the time-of-flight signal is more complicated to retrieve. The time-of-flight measurement spacing within 500 µm would satisfy most of the application requirements. For a simple design, two sensing elements should be placed downstream, and one additional sensing element could be placed upstream for flow direction identification or other functional requirements. 

### 4.2. Fabrication

Figure 3 shows the schematic presentation of the cross-sections of the basic structure of the thermal time-of-flight sensors for liquid and gas, respectively. These sensors have simple structures and easy processes and typically involve five to six photomasks/lithograph steps. There will be no special equipment required for these processes. They could be readily done by the current state-of-the-art commercial MEMS device foundries anywhere in the world with a cost that would be even quite affordable for disposable applications. For the liquid application, if the pressure required during the measurement can be high (say, more than 1 bar (15 psi), the membrane structure would have reliability issues during the measurement since the excessive pressure could lead to a deformation of the membrane resulting in a change of the spacing between the micro heater and the other sensing elements. Hence, the thermal isolation cavity could be filled with porous materials [86], or even a glass substrate could be employed. 

The overall micromachining process is very straightforward. The membrane is usually made of silicon nitride on a silicon substrate with a thickness of about 1 µm. In the process of fabrication of the microheater and sensing elements, it will be dependent on which sensing principle will be taken. For example, for thermoresistive sensing, the thermistors can be deposited via e-beam evaporation or physical vapor sputtering deposition if metal thermistors are designed. For polysilicon thermistors, various doping technology can be used. After the thermistors are patterned, the metallization process would be the next step. For most designs, a thermistor close to the substrate and upstream of the flow will also be included for measurement of the fluidic temperature such that any temperature effects of the thermistors could be compensated. The surface passivation is also often made with silicon nitride or a mixture of silicon nitride and silicon dioxide with a total thickness of less than 1 µm. After the bonding pads are opened with plasmas etching, the backside thermal isolation cavity etch process will be followed via either depth reactive ion etching or wet chemical etching. Before the sensor singulation, the surface of the sensors would now go through additional surface passivation or conforming coating process to terminate the pinhole in the surface of the top silicon nitride film. 

For microfluidic applications, some commercial approaches are to attach the chip outside a tiny thermally conductive tube; hence, a cavity would be helpful for thermal isolation and better performance. This “non-invasive” package is helpful for biological, biochemical, or other sensitive or high-pressure applications. However, the thermal barriers due to the tube wall would sometimes lead to drifting and limit the dynamic range of measurement. Although the sensor surface has the passivation layer that separates the sensing element and the liquid, there would be some challenges for the microfluidic device package in preventing the dead volume and any liquid contact with the chip’s carriers and the non-passivated areas on the chip after singulation. 

For gas measurements, direct contact with the fluids is usually not a concern. However, the membrane structure would create issues for high-pressure applications or with an abrupt gas pressure alternation. To solve this problem, the sensor membrane is often designed with some “open features” that allow the gas to quickly access the cavity underneath the membrane, leading to a quick pressure balance for the membrane. The openings are usually placed around the micro heater and the sensing elements as they can serve as additional thermal isolation. These “open membrane” designs could sometimes not be allowed in a CMOS-compatible process. Then, some package structures could be designed to allow gas access to the backside cavity. The advanced MEMS process now offers the chip VIA process that can reduce the chip size and reduce chip cost. The VIA structure will also simplify the sensor package process and enhance reliability, as the wire bonding process can be eliminated. 

### 4.3. Device Operation and Data Process

Figure 4 shows the block diagram for the basic components of a complete functional thermal time-of-flight sensor. Compared with the other thermal sensors, the critical difference for the thermal time-of-flight sensor is that the microheater is driven by modulated heat, and measurement from the sensor would require both time domain data and analog data (amplitude or temperature deviations). For heater modulation, the most frequently used approaches in the literature are pulse, square wave, and sinusoidal wave. With the current electronics, such modulation is very easy to realize via an MCU and an amplifier. In some MCUs, amplifiers are already integrated; hence, one high-performance MCU alone can create the modulation. The pulsed or square wave (extended pulse becomes DC when pulse time is further extended) would be simple. However, a sinusoidal wave will generate much better resolution or yield better measurement accuracy. The measured “time-of-flight” from a micromachined sensor would be in microseconds. The data stream would require the MCU to have a better source for the data process than a typical calorimetric measurement. The time-of-flight or phase shift can also be measured via a pure hardware demodulator. Unless only analog output is required, the digital data process will also require high-resolution ADCs of at least 16 bits to have the desired accuracy. To maximize the benefit of the thermal time-of-flight sensing technology, data should be acquired from at least two sensing elements for multiparameter acquisition and fluidic property-independent measurement. Similar to all thermal sensing technology, measurement of the fluidic temperature is also very important for the temperature compensation of the control scheme. The typical frequency spectrum of heater response and signals acquired from these sensing elements are shown in Figure 5, where a 100 Hz driving sinusoidal modulation was applied, and the sensor was a silicon nitride membrane-based gas sensor [57]. It should be noted that the driving frequency for the micro heater should not be close to the harmonics of local city electricity. These wanted harmonics can be filtered out easily with demodulation applied in the data measurement.

Selecting a proper driving frequency would be essential, depending on the applications and sensor design. A fast-driving frequency may have less interference, but the signal-to-noise ratio could not be satisfied. It would be helpful to perform a frequency scan before finalizing the circuitry design to determine the optimized frequency. It would be instrumental in designing a single sensor for applications of multi-fluidic or mixed fluidic measurements [59]. For a general-purpose measurement, a frequency below 100 Hz would be recommended, with the exclusion of those applied for local city electricity, which will inevitably be shown in the spectra but can be easily filtered out in the data processing. The data processing and subsequent calculation of the phase shift or the corresponding heat transfer time from the flowing fluid at a specific speed could be done with either a hardware demodulator or software. Some commercially available pre-phase-lag detectors with precision lock-in amplification technology can be used for the signal conditioning circuitry. The time domain data is usually more stable and have less drift compared to analog calorimetric or anemometric temperature data acquisition scheme. 

Figure 6 shows the acquired data from the calibration of a time-of-flight sensor for microfluidic applications. The sensor had a microheater and two thermistors as the sensing elements placed at a distance of 260 and 110 µm from the microheater. All the elements were made of platinum with a linewidth of 4 µm. The microfluidic channel had a cross-section of 2.0 × 0.5 mm. The sensor was calibrated with purified water using a high-precision syringe pump together with a high-precision balance. Detailed information can be found in a previous paper [67]. The output phase shifts were plotted in a polar plot in Figure 6a. The reference flow rate against the phase shifts is shown in Figure 6b. The polar plat would provide important information for the performance of the sensor, and they are a direct presentation of the relative phase shift and the smoothness of the data output (calibration data acquisition errors or any system errors). It could also help the examination of the offset information, particularly for multiple sensing elements. The relative shifts would be more straightforward for visual scrutinization. On the other hand, the calibration curve would be critical for the data processing and the device’s accuracy performance. The calibration curve shows a non-monotonic transition at the low flow rate regime where thermal diffusion dominates. In the theoretical approximation, the offset would be a measure of the fluidic diffusivity, as shown in Equation (3). If the theory holds, the offset should be a constant independent of the relative distances between the sensing elements and the micro heater. The measured data shown in Figure 6b, however, indicated the offset strongly depends on such a distance, and the closer distance has a smaller offset. The results suggested that there would be multiple heat transfer paths in the practical case. For the data presented in Figure 6, the thermistors were made on the silicon nitride membrane. Since silicon nitride’s thermal diffusivity is larger than that of water [87], the heat transfer process would be a combined heat transfer via both water and silicon nitride. These differences could explain that the observed data that has a larger space between the microheater and the sensing elements would have a bigger offset or effective diffusivity using the two-phase heat transfer model [88]. The results could also be used as a guideline for the sensing element spacing design for applications if the thermal property measurement is also needed. The transition from the diffusion-dominated regime to the time-of-flight regime could also be estimated with the model discussed in the literature [75]. Within this regime, the “effective time” increased with the flow speed, as it would be the three or four heat transfer paths lapped together, i.e., diffusion in the fluid, diffusion in the substrate, flow speed contributions, and possibly some effects from the diffusion at the channel walls, as in the case of microfluidic applications.

Thermal time-of-flight sensors can obtain calorimetric or anemometric data and time domain data at the same time. The microheater and sensing elements’ temperature differences or amplitude changes can be measured using the classic calorimetric or anemometric sensing circuitry. Figure 7 shows the calorimetric data (a) and time-of-flight data (b) from the same sensor discussed above. The fact that the calorimetric data can also be measured is undoubtedly an advantage over pure calorimetric sensing as the additional data would provide much more information, including mass flowrate, flow speed, or volumetric flow rate via a calibration procedure. Additionally, the measured time can be used to calculate the flow speed since the distances are well-defined. This feature can potentially be used for the development of the self-calibration scheme. At zero flow, the thermal diffusivity can be measured as discussed above, while thermal conductivity could also be directly measured using the microheater’s thermal (power) consumption value. With all these data, the measurement accuracy could be cross-referenced and supported for monitoring the sensor reliability performance. A well-designed thermal time-of-flight sensor would require a careful examination of all these factors, as the spacings and numbers of the sensing elements would be all critical for the data being acquired. For example, the spacings between the microheater and sensing elements on a micromachined thermal time-of-flight sensor usually are much larger than those used for calorimetric or anemometric sensors. The thermal time-of-flight signal will be more pronounced outside the diffusivity regime. The smaller spacing for calorimetric sensing will allow a better signal-to-noise ratio and data linearity. Therefore, in this respect, thermal time-of-flight sensing would require a low flow rate compensation. This issue needs to be solved at the device level design. If the low flow speed cannot be deconvoluted from the diffusion-dominated signal, a calorimetric element could be added to gain the dynamic range. This aspect will be further addressed in the following section for applications. 

One advantage of thermal time-of-flight sensing is that the data can be fitted with an empirical formula based on the theoretical models. For the data shown in Figure 7, the fitting from these two sensing elements could be done with a single formula, close to that proposed in [66].
(8)$$f=A+B/\left({\left(t+C\right)}^{D}+E\right)$$
where *A*, *B*, *C*, *D*, and *E* are fitting constants.

The empirical formula would allow the measurement to be more accurate than the calorimetric or anemometric measurement, where the data linearization would not usually be done via an analytical model.

## 5. Applications

Although the micromachined flow sensor was one of the earlier examples of successfully commercialized micromachined sensors, unlike the others, such as pressure sensors and accelerators, high-volume application has been relatively limited to a particular automotive application. On the one hand, flow applications involve complicated fluidic dynamics and control electronics; on the other hand, calorimetric or anemometric sensing has certain limitations. Many potential applications are being held back. These limitations include fluidic composition dependence, power consumption, package, and materials compatibility. Microfluidics has been thought to be a killer application for micromachined flow sensors in the past two decades, but the actual progress is very limited. Utility gas meter applications also have been discussed for over 20 years, but the critical mass is yet to be achieved. The advantages of thermal time-of-flight technology would provide some remedies for the missing driving force, such as simultaneously offering fluidic composition independent flow and fluidic thermal property measurement. Nevertheless, practically it still requires efforts for its success in commercialization. This section discusses some potential “killer” applications. It is believed that further developed thermal time-of-flight flow sensing technology might facilitate future growth in these fields.

### 5.1. City Utility Gas Metering

The utility gas meter market will be about USD 4 billion in 2022. Mechanical flow measurement technology is still dominant in today’s utility gas metering and is considered one of the most successful application technology in history. However, the advancement of this technology is also the slowest one from a technical point of view. More than 90% of residential flow meters are still made with the same mechanical diaphragm approach as 170 years ago. The slow technology evolution could be partially due to the meters being used for tariffs. Utility gas is one of the essential supplies for daily human life, which also involves heavy government regulation, and any changes will lead to lengthy processes. On the other hand, the meters also pose challenges that cannot be easily managed even with today’s technical capabilities. Utility gases include natural gas, liquefied petroleum gas (LPG), biogas, and others. These gases have complicated gas compositions. The meters, once installed, must be running for a minimum of 10 years without any maintenance, and in many countries, such a meter can even run for over 30 years without any interruptions. External power is often not readily available and cannot be an option. Photonic power also suffers reliability such as dust attacks and problems with tamper resistance, even the long-life battery would be questioned by the users and the promised power life is still not sufficient. Even though the diaphragm meter only offers volumetric gas measurements and low accuracy, trade deficits were always an issue with gas distributors. The current meters’ mechanical power and maintenance-free features still make them difficult to be replaced. In 1998, the US DOE (Department of Energy) sponsored a project to identify alternative approaches to meter utility gases with the ultimate goal that the meter should measure energy instead of volume. However, the year-long studies concluded that none was readily applicable to the targeted energy metering capability [89]. The technological evolution and advancement in utility gas meters have been summarized in a previous publication [25]. Here are the updates in recent years and opportunities for thermal time-of-flight technology.

In the past few years, utility meters made with micromachined calorimetric mass flow sensors have competed with ultrasonic utility gas meters. The advantages of ultrasonic meters are that it measures the volumetric flow rate that will not conflict with the existing metrology standards. It has been used in custody transfer at the gas distribution station at which the multi-channel ultrasonic technology and temperature and pressure compensation are applied. In contrast, the attractiveness of the micromachined thermal calorimetric flow sensing technology for gas distributors is that it offers mass flow rate measurement. The mass flow rate is preferred over the volumetric flow rate and is close to the final desired energy metering. The prototype of a micromachined thermal mass flow utility meter was built in 2002 for European residential applications [90], but commercial and industrial applications commercialization took place earlier [91,92]. Some research efforts for improving sensor design and even capacitive sensing approaches were reported for a more cost-effective or performance enhancement in natural gas flow applications [93,94]. In 2016, Italy published its national standard for micromachined thermal mass flow meters [95], and China published a broader national standard for this technology in 2017. These standards open the market for the rapid deployment of the technology. More than 3 million utility gas meters with micromachined thermal mass flow sensors have been installed in quite a few countries. In 2021, the European Committee of Standards published EN 17526 [96], which further facilitates market acceptance of the technology and fuels its growth. Nevertheless, there are still many concerns about the utility industry’s applications of micromachined calorimetric sensors. One of the critical issues is the gas composition dependent and compensation schemes, although the published standards allow some additional margins for gas property deviations in metrology. Another issue is long-term metrology reliability and the ultimate goal of energy metering. The current compensation approach to the metrology deviation due to gas composition changes is to acquire thermal conductivity data with an additional thermistor or some approach to take the conductivity data via the flow speed plateau [97]. The effectiveness of such compensation for metrology could not be well attested due to the large variety of gas compositions. The evidence was most from the gas groups specified by the European utility gas standard [98].

The attractiveness of thermal time-of-flight sensing is to have multiple sensing elements that could, in theory, remove the fluidic property sensitivity during the flow measurement. There would be additional advantages that the micromachined sensors could easily integrate a plural number of sensing elements, measure thermal response, and arrange precise spacing. In some earlier examples using either micrometers (8 µm for the heater and 3 µm for sensing elements), coplanar wires [99], or micromachines sensors [51], the data showed that the fluid-independent measurements could only be achieved in a very limited dynamic range. The accuracy seemed unsatisfactory if a custody transfer standard was to apply. Figure 8 shows the flowrate measurement data of two gases (air and methane) from a micromachined thermal time-of-flight sensor. Figure 8a is the raw calibration data (air) and then direct measured with methane (b) without a gas conversion factor or any thermal conductivity compensation algorithm. The calibration and calculation algorithm was based on Equation (7). Figure 8b shows the metrology accuracy of the measurements. The sensor was made into a flow meter with a flow channel of 3 mm in diameter. The meter was first calibrated with the air, and the gas was changed to methane for direct measurement. For the micromachined calorimetric or anemometric sensor, or a classical capillary calorimetric sensor, a gas conversion factor will be required to measure methane if the meter is calibrated with air. The factor was around 0.65, depending on the actual flow meter design. In Figure 8, one could observe that the thermal time-of-flight technology is effective for a reasonable metrology performance. However, it is also clear that the deviation became undesirable towards the low flow rate, where the diffusivities and convection would compete. Hence, additional data processing would be needed for better performance in the full dynamic range. One simple yet effective remedy to the low flow rate deviations is to promote effective measurement speed. Figure 9 shows two such approaches. Figure 9a is a sketch from the proposed hotwire thermal time-of-flight flow meter [38,100], and (b) is a configuration with a micromachined thermal time-of-flight meter [65]. Both are designed for city utility gas metering applications. The basic concept is to accelerate the low flow speed at the measurement point such that the diffusivity regime could be minimized. The experimental data were very encouraging as they also offered a direct internal monitor of the sensor performance. The data shown in Figure 8 for the discussed sensing scheme was from measured time and then calculated using Equation (7). Simultaneously, another calibration using Equation (8), which is from a single sensing element, could also be established. These two would have different deviations when any contamination-related performance occurs. By comparing these two sets of data, one could predict the sensor status, and various algorithms could be applied for compensation even via the remote cloud data. Subsequently, this algorithm is surely required or subject to any regulatory terms or standards. 

The ultimate goal of energy metering for utility gas applications is highly challenging for the required cost and performance. The current approaches for natural gas calorimetry are either by gas chromatography or a complicated calorimeter [101]. Some studies have addressed this issue with a micromachined sensor [102,103] and a highly integrated calorimetric, pressure, and Coriolis MEMS sensor [104]. However, cost and reliability are still concerns. The basic requirements for a successful yet simple energy metering sensor for utility applications would have the capability to measure gas composition, calorific value, and density, in addition to flow measurement. A thermal time-of-flight sensor would be applicable for these measurements [25]. However, due to the broad spectrum of the gas compositions, this technology would also be inferential, and additional studies would be required to document the applicability. In addition to the functionality, power consumption would be another challenge. Unlike a micromachined calorimetric flow sensor with a clear pathway for power saving with further miniaturization and linewidth reduction, a thermal time-of-flight sensor would intrinsically require higher power such that it can acquire all desired parameters. Efforts for smart energy harvesting devices could be required to combine with technology to meet the ultimate requirements of the utility industry and for abrupt growth.

### 5.2. Medical Applications

Medical applications would be another area where the technology could offer innovative solutions. Micromachined calorimetric sensors have been widely used in medical ventilators, endoscopes, asthma detection, cancer plasma treatment, and lung function recovery equipment, to name a few, in addition to applications in continuous positive airway pressure (CPAP) therapy. However, the volume of medical applications is relatively low except for CPAP application. Medical oxygen therapy is a very old medical treatment tracing back to 1798 [105]. However, the current supply of oxygen in hospitals or in homecare is controlled by mechanical rotameters, which can have significant errors [106], regardless of where these devices are made. In recent years, chronic obstructive pulmonary disease (COPD) is becoming one of the top public diseases worldwide [107], with over 400 million patients in 2019, causing a few million death each year. In particular, due to the impact of COVID-19, homecare oxygen concentrators have grown rapidly. Oxygen therapy, as one of the treatments for these diseases, has now attracted quite some attention for the refinement of the treatment technologies. 

Oxygen therapy includes a controllable precise oxygen delivery rate and instantly measured oxygen concentrations. The current technologies for these measurements are, however, far from satisfactory. Mechanical rotameters have errors in flow rate measurement. They are also unable to transmit data remotely, and oxygen concentration measurement largely depends on electrochemical or ultrasonic measurements. The electrochemical sensors have a very slow response time of 15+ s [108]. The ultrasonic oxygen concentration measurement has considerable uncertainties as it is an inferential detection depending on pressure and temperature data. Nevertheless, low-cost oxygen concentrators often lack temperature and pressure measurements. Temperature will impact the oxygen volume measurement accuracy and the precise distance of the ultrasonic transducers, adding to an already big measurement uncertainty [109]. Figure 10 shows a set of oxygen concentration data measured by a thermal time-of-flight sensor based on the theoretical understanding that the effective diffusivity of a fluid depends on its concentration. For the binary fluidic mixture (in this case, it was a mixture of oxygen and nitrogen), measured diffusivities correlate to the oxygen concentration. To decouple the influence of flow during the measurement, the sensor was placed slightly inside the flow channel wall where the dynamic flow speed would be null. The data shows an excellent monotonic correlation in Figure 10a. The response time in this data set was about 20 ms with the designed control electronics. To test the flow influence for the measurements, the ambient air with about 21% oxygen was used for the experiments. The acquired data were then used to calculate the diffusivities using Equation (6). The calculated results are shown in Figure 10b, and a strong dependence on flow rate was observed. Hence, in practical cases, the diffusivity measurement should be confined within the no-flow space for an easier data process. The detailed analyses of such influences need to be further explored, but likely in the actual configurations, the thermal transfer might involve multiple paths. The results shown in Figure 10 are certainly encouraging as the technology offers a very low cost while having the flow and concentration data measured simultaneously. Since this approach is a direct measurement of the physical parameter or property (diffusivity), and it has fewer effects by pressure and temperature, it could potentially provide desired replacements of the current technologies for oxygen concentration measurement in medical oxygen therapy applications. Additional medical applications such as respiratory analysis for carbon dioxide and anesthesia gas concentration control could also be applied. 

### 5.3. Microfluidics

Microfluidics has been one of the fast-growing devices in the MEMS industry because of its wide applications in biotechnology, medical equipment, pharmaceutics, and others. Among all research frontiers, genomics, point-of-care diagnostics, and drug delivery are driving market growth with quite some success in commercialization. Point-of-care and drug delivery are particularly interesting for thermal time-of-flight sensing approaches. Point-of-care diagnostics allow fast processing of a small bio-sample and enable self-diagnosis for an aging society in many countries [110]. Drug delivery aims for more precise drug infusion and prevention in handling mistakes [111,112]. These two areas require billions of devices if the desired functions can be achieved. The traditional microfluidic flow measurements using Coriolis or thermal capillary devices are very costly and bulky and are only used in laboratories. Micromachined commercial thermal liquid microfluidic flow sensors have emerged in the last decade. These commercial products utilize different thermal sensing principles that cover the three major technologies with thermal calorimetry, anemometry, and thermal time-of-flight, and some efforts of making micromachined and commercially available Coriolis sensors were also reported [12]. However, the complicated microfluidic process and the making process of the devices have greatly hindered the growth of flow-sensing products for microfluidics [113]. Besides the high commercial costs, the calorimetric and anemometric flow sensors require calibration with real fluid for desired precision or metrological accuracy. Microfluidic properties often have a nonlinear response in the full dynamic range of thermal sensing, which makes the calibration with common fluids unrealistic. The limited dynamic range and accuracy are not desirable for the precision requirements of many microfluidic applications such as drug infusion. For practical reasons, manufacturers would be unable to offer real fluidic calibration either because of small volume demands or the availability of uncommon fluids. This is similar to the current calibration option for anesthesia gas sensing with micromachined calorimetric flow sensors where carbon dioxide is used for calibration. Even though these two gases have quite close thermal properties, nonlinearity and deviations are always questioned. For microfluidics, more physical or even chemical factors will impact fluidic metrology [12]. If flow sensing products could only provide the mass flow rate measurements under certain calibration conditions, that will surely not be appreciated by the applications. Additional fluidic information such as fluidic concentration and physical or even chemical properties of the fluids are often required at the same time. Thermal time-of-flight sensing would offer a good opportunity for these applications [114], and further integration would make the sensor more powerful for understanding and controlling microfluidic applications. Figure 11 shows that a thermal time-of-flight sensor could offer a better solution than calorimetric sensors. The sensor was made with one microheater and two sensing elements. The chip was passivated with silicon nitride only without further surface coating for pinhole termination. Silicon nitride surface inhibits diffusion of water and oxygen if no passivation is applied post-processing [115]. To test the long-term surface stability against water, the bare sensor chip was subject to surface contact angle measurement with an interfacial tensiometer. After the first measurement of the dry chip surface, the chip was immersed in de-ionized water for 24 h. Then, it was taken out of the water and dried with a nitrogen gas gun until no water could be seen under visual inspection. The water contact angle was measured again with the same procedure. It was observed that the surface contact angle would gradually decrease from a near hydrophilic of 32° to about 21° during a continuous daily measurement for 5 days. A new chip from the same location on the same wafer was packaged into a microfluidic channel with a square-shaped cross-section of 1.5 mm × 0.75 mm. The channel material is PEEK. The sensor was then subject to calibration with de-ionized water and verification with a precision syringe pump and a precision electronic balance. The same verification procedure was performed after the sensor was immersed in the de-ionized water for 24 h, and the accuracy measurement was performed sequentially for 5 days. Figure 11a compares the data for days 1 and day 5. It could be observed that with the same delivered flow rate via the syringe pump, the sensor registered a substantial negative deviation for the measurement performed on day 5. This deviation could be explained by the data measured from the tensiometer, as the surface tension would alter the flow profile and change the thermal response of the sensing element beneath the surface silicon nitride film leading to the deviations in metrological data. The polar plots showed that the deviation was constant against the flow rate after a certain flow speed. The same reason would be applied to the different deviations toward a lower flow rate, where the wetted surface will lead to the alternation of the surface thermal diffusivity. This phenomenon has also been reported by others for micromachined calorimetric microfluidic flow sensors [116]. It was further found that if the sensor’s power was kept on, the change of the surface tensions could be accelerated. This metrology instability would be a very challenging issue for the flow rate measurements using the thermal flow sensing principles. Fortunately, with the dual sensing elements, this reliability issue for the measurement could be eliminated, as shown in Figure 11b, where the polar values were obtained by taking the relative phase shifts of the two sensing elements. For the same sensor, the relative shifts did not have deviations when the diffusion regime was excluded, while for the lower flow range, such effects were also reduced. Therefore, by simultaneously acquiring all these data and applying the necessary analytical data process, the measurement could retain its accuracy. At the same time, the changes in physical conditions could also be captured.

The thermal time-of-flight sensing approaches also offer the substantial advantage of the fluidic composition independent measurement that would allow a water-calibrated sensor to be applied for other liquids without additional adjustment or compensation. However, achieving such an objective in the fully dynamic range would be difficult as the low flow rate regime was overtaken by diffusivity. It would require additional work to decouple the complicated physical process before better results could be obtained. The current data showed that the micromachined sensor works well in a 10:1 range for a composition-independent measurement, and this range would be acceptable for some applications. With this possibility, the products will substantially reduce the cross-contamination and post-calibration costs for many biotechnical and medical applications. Further, the technology would be capable of offering the fluidic property data and have less influence by the environmental temperature variations. The most favorable capability of self-calibration would require full and precise automation in the product assembly as the dimension of the sensing elements would have a very high requirement for consistency. Currently, it will still require the initial calibration of the instrumental factors.

### 5.4. Other Applications

In addition to the above-discussed potential killer applications, there are many other applications for which thermal time-of-flight sensors could offer a solution that current thermal flow sensors could not provide. For example, appliance efficiency such as refrigerants or heat pumps for air conditioning would need a better controller flow metering system. A recently reported thermal time-of-flight sensor using a small heater and two thermal couples for this purpose has shown limited success [117]. In this case, a micromachined sensor could be a better option with respect to performance as well as cost. The market size would also be very great if the product were designed correctly. Similarly, for an industrial or home refrigerator, coolant flow rate and concentration would be the two parameters that can assist in gauging the efficiency and allow the monitoring of the coolant degradation preventing reliability issues. It could further offer substantial energy savings for environmental protection demands. For automobiles built with diesel engines, control of the exhaust process requires the precise measurement of the blue fluid in which the urea concentration is critical for the emission, in addition to the flow rate measurement. At the moment, the ultrasonic approach is used, but similar to the oxygen concentration measurement, alternative technology has been discussed for years. For welding gases, a cost-effective device is required for monitoring both the flow and the ratio of the gas mixtures. Another application is beer or wine fermentation process control, where the carbon dioxide release and concentration directly impact the final product quality. Combustion efficiency also demands a better measurement capability for both gas and gas concentration or energy in many home appliances and industrial process control.

## 6. Concluding Remarks

Among the micromachined commercial thermal sensing devices, thermal time-of-flight is less popular. One reason could be rooted in the fact that the early commercially successful thermal flow sensing products were either made with hotwire anemometry or capillary calorimetry. The traditional hot wire thermal time-of-flight sensor does not have any competitive advantage. It is still fluidic composition-dependent. Further, it suffers from slow response, low flow speed insensitivity, and small dynamic range. It also has more troubles during installation, and the tiny wire does not guarantee reliability for many applications. The only attractive feature would be its capability in turbulence flow measurement. The electronics were less advanced when the micromachined thermal flow sensor was commercialized in the 1980s. The earlier products were mostly made with analog circuitries. All these factors would not encourage the commercialization of thermal time-of-flight sensing products. However, today’s electronics have no comparison with those in the 1980s, and they offer many more options for the signal process at a very low cost. Furthermore, the micromachining process capability and package advancement could change the landscape for the technology. The current issues with commercial thermal flow sensing products have also been a driving force in identifying pathways for innovation of the technology.

Since the micromachined thermal sensing technology was incepted, the research activities on the thermal time-of-flight sensing approaches have been far less than those for calorimetric and anemometric sensing technologies. As we have discussed above, many drawbacks of classical thermal time-of-flight sensing could be eliminated with the advanced micromachining process and state-of-the-art electronics. These include fluidic property independent measurement, simultaneous fluidic thermal property or relative fluidic concentration data, small signal capability, and higher accuracy in turbulence flow. In particular, the possibility of self-calibration and self-diagnosis for reliability can not only substantially reduce costs but would also be indispensable for some biotech and medical applications. Compared to the current calorimetric or anemometric micromachined sensors, reliability is another factor that thermal time-of-flight sensor could offer some additional benefits, although the weakness of a membrane design could not be performing in a fluid that is not clean and has abrupt external pressure changes or other situations that could damage the membrane. Thermal time-of-flight can measure both the time domain and amplitude or calorimetric data which could be used for cross-correlation when the sensing element surfaces have light contamination or even surface deposition. If the changes in surface conditions take place, the thermal time-of-flight sensor could send the alarm or cancel out such changes with signals from multiple sensing elements when such changes are not extremely localized. This cross-correlation feature is very helpful for sensor reliability, in particular for long-term performance.

Among the foreseeable disadvantages, power consumption could be a major challenge as the algorithm will require much more computing power to obtain the desired results, and the thermal time-of-flight sensors might not readily adapt to some of the current effective low-power designs in calorimetric sensors. Another disadvantage is that thermal time-of-flight sensing intrinsically has a smaller dynamic range, and more physical parameters are involved in the data processing. However, these could be amended via additional sensing elements and a more intelligent flow channel design. More work is surely needed to explore the low flow speed regime so that better performance can be made available for some applications. It is believed that with more effort, thermal time-of-flight sensing technology could lead the abrupt growth in various applications of micromachined thermal sensing devices.

## Figures and Tables

**Figure 1 micromachines-13-01729-f001:**
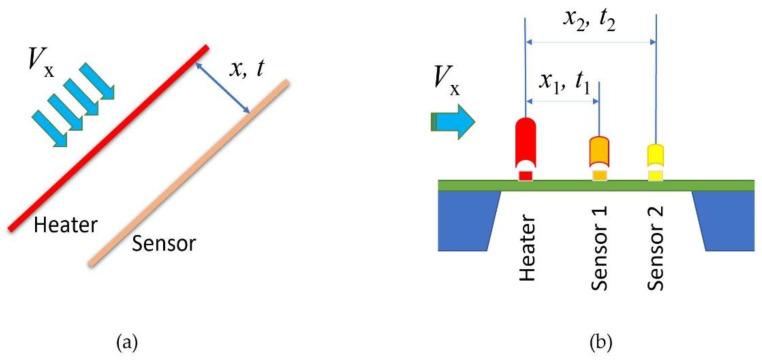
Schematic for the thermal time-of-flight sensors: (**a**) classic pulsed wire and (**b**) micro- machined sensor.

**Figure 2 micromachines-13-01729-f002:**
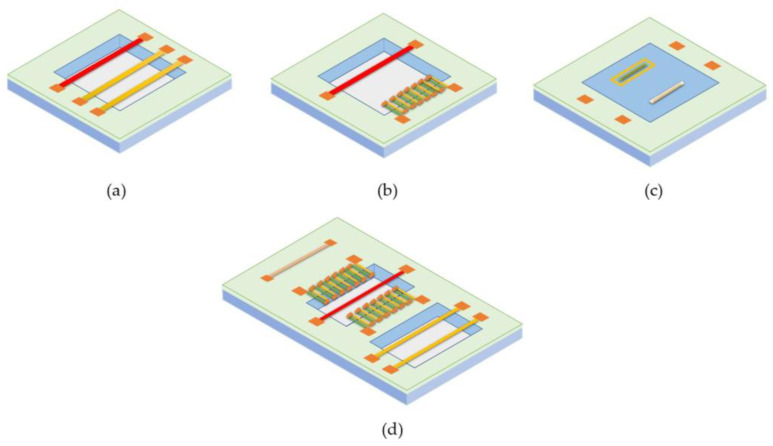
Schematic for the thermal time-of-flight sensors: (**a**) thermoresistive, (**b**) thermoelectric, (**c**) thermoelectronic sensing configuration, and (**d**) a fully range sensor.

**Figure 3 micromachines-13-01729-f003:**
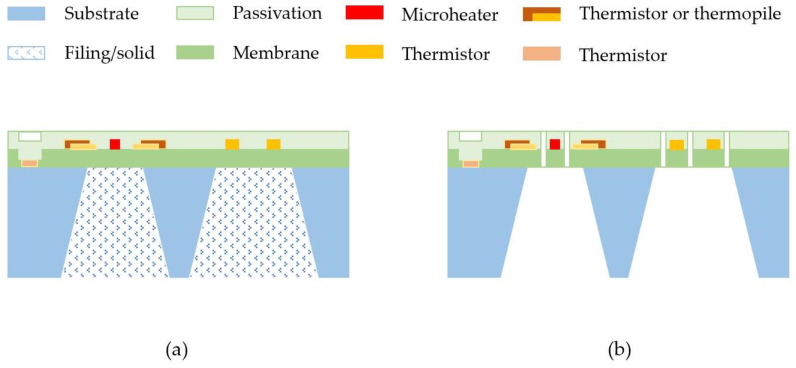
Schematic of the thermal time-of-flight sensor cross section showing the basic structure layers: (**a**) liquid sensor, and (**b**) gas sensor.

**Figure 4 micromachines-13-01729-f004:**
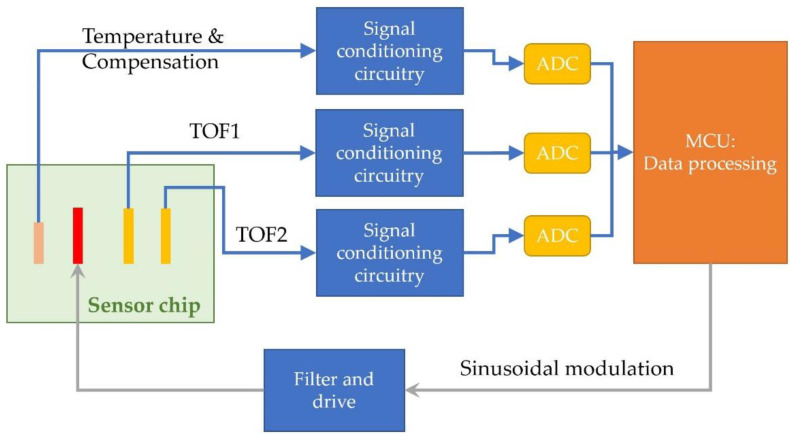
The block diagram for the control electronics for a thermal time-of-flight sensor.

**Figure 5 micromachines-13-01729-f005:**
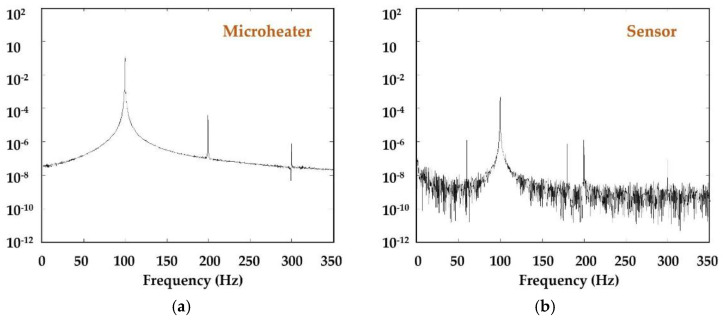
The typical frequency spectra for the microheater (**a**) and a sensing element (**b**), from a micromachined gas sensor with silicon nitride membrane.

**Figure 6 micromachines-13-01729-f006:**
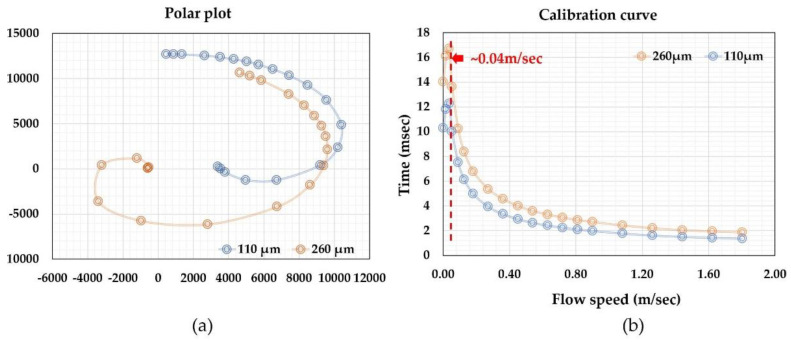
Polar plot (**a**) and phase shift vs. flowrate curve (**b**) from the calibration of a thermal time-of-flight sensor applied for a microfluidic (water) measurement from 0 to 1.8 m/s.

**Figure 7 micromachines-13-01729-f007:**
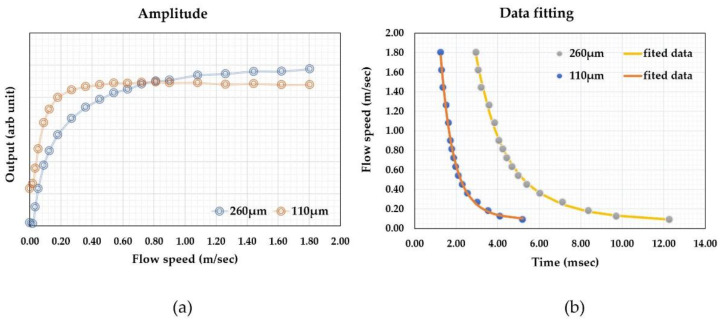
Simultaneously acquired data for (**a**) amplitude (calorimetry) and (**b**) thermal time-of-flight data from the microfluidic sensor discussed above.

**Figure 8 micromachines-13-01729-f008:**
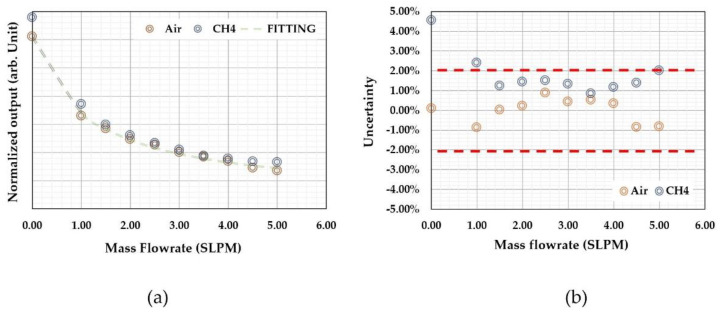
Gas composition independent measurement data with the micromachined thermal time-of-flight sensor: (**a**) is an output calculated using Equation (7) and additional compensation, and (**b**) is the metrology performance of the meter.

**Figure 9 micromachines-13-01729-f009:**
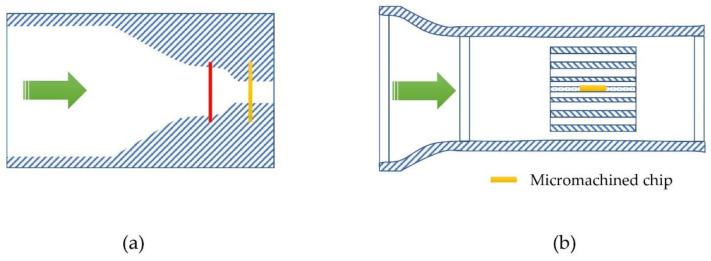
Configuration of a thermal time-of-flight sensor inside a flow channel: (**a**) schematic from [36] for a proposed natural gas meter with two hot wires; (**b**) gas meter flow module with a micromachined thermal time-of-flight sensor.

**Figure 10 micromachines-13-01729-f010:**
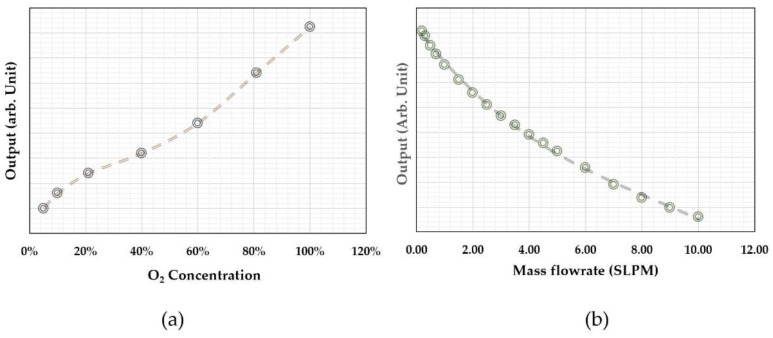
(**a**) Measured oxygen concentration for oxygen and nitrogen mixture in the static conditions and (**b**) the calculated diffusivity against the mass flow with 21% oxygen (ambient air).

**Figure 11 micromachines-13-01729-f011:**
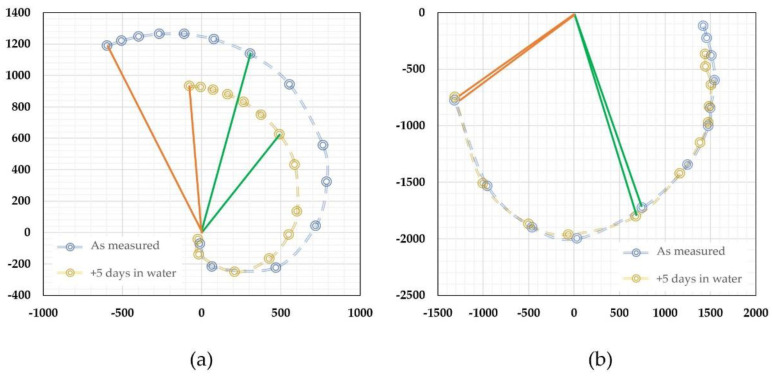
Example of a thermal time-of-flight sensor for compensation for long-term reliability and additional property sensing: (**a**) Sequential measurement of flow rate in a wetted flow channel with changing surface tension; and (**b**) removal of the surface tension effects with multiple sensing elements. The green lines and brown lines correspond to the same flow rate.

**Table 1 micromachines-13-01729-t001:** Micromachined thermal time-of-flight sensors in literature.

Medium	Materials/Principle	Excitation	Range	References
Gas (air)	Pb/PbTe Thermoresistive	Sinusoidal	0~250 g/s	1986 [40]
Gas (air)	Doped silicon Thermoelectronic	Square wave	2~30 m/s	1988 [42]
Liquid	Doped silicon Thermoelectronic	Square wave	0~12 mL/min	1991 [43]
Liquid	Doped silicon Thermoelectronic	Square wave	0~10 mL/min	1992 [44]
Liquid/Gas	Pt/Ti Thermoresistive	Pulse	0~5 m/s	1994 [45]
Gas	PolySi Thermoelectronic	Sinusoidal	0~30 m/s	1994 [46]
Gas	Pt Thermoresistive	Pulse	0~20 mL/min	1995 [47]
Gas	Pt Thermoresistive	Pulse	0~0.08 m/s	1995 [39]
Liquid/Gas	PolySi/thermopile Thermoelectric	Pulse	0~0.08 m/s	1999 [48]
Liquid	Pt/Ni/PZT Thermoelectric	Pulse	0~0.25 m/s	2000 [49]
Liquid	Doped silicon Thermoelectronic	Pulse	0~1 m/s	2002 [50]
Liquid	Pt/Parylene Thermoresistive	Single pulse	0~30 µL/min	2003 [51]
Liquid/Gas	PolySi Thermoelectronic	Sinusoidal	0~0.025 m/s	2003 [52]
Gas	Pt Thermoresistive	Pulse	Unspecified speed	2006 [53]
Liquid	Cr/Ge Thermoelectric	Sinusoidal	Thermal properties	2006 [54]
Gas	Cr/Ge Thermoelectric	Sinusoidal	Thermal properties	2008 [55]
Gas	Cr/Ge Thermoelectric	Pulse	0~1.4 m/s	2009 [56]
Liquid/Gas	Pt Thermoresistive	Sinusoidal	0~30 (0~0.06) m/s	2010 [57]
Liquid	Pt Thermoresistive	Pulse	0~1 m/s	2011 [58]
Gas	Si/thermopile Thermoelectric	Sinusoidal	Thermal properties	2011 [59]
Gas	Pt Thermoresistive	Pulse	0~300 mL/min	2012 [60]
Gas	Doped silicon Thermoelectronic	No data	0~2 m/s	2014 [61]
Gas	Cr/Ge Thermoelectric	Sinusoidal	0~2 m/s Thermal properties	2014 [62]
Liquid	Ag/thermocouple Thermoelectric	Sinusoidal	0~70 µL/min	2018 [63]
Gas	Au/Ni Thermoresistive	Square wave	0~7 m/s	2019 [64]
Gas	Pt Thermoresistive	Sinusoidal	0~30 m/s	2019 [65]
Liquid	PtAu/NTC Thermoresistive	Pulse	0~1500 µL/min	2021 [66]
Liquid	Pt Thermoresistive	Sinusoidal	0~50 mL/min	2020 [67]

## Data Availability

Not applicable.

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
