# Peer review of "Micromachined Thermal Time-of-Flight Flow Sensors and Their Applications"

_micromachines, 2022, doi:10.3390/mi13101729_

Round 1
Reviewer 1 Report
1. Page 2 line 95, what are the additional parameters? The author should give a detailed explanation of this content.
2. the logic of the whole draft is not that good, sometimes the reader cannot grasp the critical point that the author wants to express.
Author Response
Thank you for your comments.
- I have added contents to detail the parameters discussed.
- I tried to add additional explanations at the introduction for the logic of the paper.
Reviewer 2 Report
I congratulate the author for putting together a good review article on MEMS thermal mass flow sensors. The article takes care of major aspects of the review article like introductory theory, an overview of the present scenario, and extensive future prospects.
I have a few minor comments on the manuscript.
1) Throughout the manuscript please maintain one style of numbers with units, (for example either 1 mm or 1mm.) I observe both styles, consistency would help.
2) Line 930, Adding a reference might help as exact numbers are mentioned.
Author Response
Thank you for your comments.
- Thank you for the observation. I have made changes to keep the consistency of unit style. A total of 42 changes were made.
- A reference is added, [109]
Reviewer 3 Report
In this review article the author introduces the thermal time of flight micromachined flow sensors and reviews the literature on these sensors. He reviews the advantages and disadvantages of these sensors compared to more conventional calorimetric MEMS sensors. The review is thorough and every references on this subject is included in the manuscript. The MS can be accepted after addressing the following suggestions,
- Given that the review articles are for more of general audiences than people who are working in this field, I suggest through a simple cartoon, author tries to explain the working principles of thermal ToF operation (in the beginning of the MS) in addition to the already good explanation through equations in the body of the MS.
- I believe some of the same drawbacks that hinders the wide spread use of calorimetric sensors also exists in ToF sensors as well (e.g. reliability). I suggest the author to clearly explain why calorimetric sensors might be more susceptible to these drawbacks.
- The author needs to provide a direct reference for many of the claims in the body of this MS (e.g. l. 35 43 47 71 and etc)
Author Response
Thank you for your comments.
- A cartoon is added as Figure 1.
- Added the remarks for reliability comparison to current calorimetric products in the last section.
- Add 3 references 13-15. My understanding is that the comments are mostly for market data in the introduction section. Some observations stated are based on author's correlations with customers in the past 15+ years, published references would not be available for all observations.
Reviewer 4 Report
Generally, research has a great impact on the state of the art, which will allow the development of new knowledge and technologies. However, it seems relevant to me to add and respond to the following recommendations:
1.A table could be included as a summary of the most important parameters of the devices, and their characteristic images about micromachined or applications.
2.There is no continuity from reference 40 to 41 in table 1, check.
3.Reference 92 is not found in the text, check.
4.The declared literatures should be verified, the years from the least to the most frequent appearance.
5.The reference in Figure 2 is not observed, or is it self-created?
6.Improve image quality, mainly Figure 4-7, 9,10.
7.Describe the variables in equations 5, 6, 7 and 8. For example, ¿x = x0 = x1 = x2?, ¿t=t1=t2? and others, verify.
Author Response
Thank you for your comments.
- It could be a bit difficult to add a table as generic specifications are not easy to cover the diversity of applications and product design. However, this comment is very relevant, I tried to add a paragraph to summarize the key parameters for the device design in Sec 4.1.
- Double checked. This is because some references are not related to data in Table 1 but cited in the text.
- Thank you. This is corrected. 92 was mistaken to 94. In the revised manuscript, because two new references are added, the number now remains 94.
- Thank you. The sources are verified, and corrected.
- The figure is self-created to illustrate the structure.
- Figure 5(4),6(5),7(6),8(7), and 9 (10), 10(11) are replotted with better color.
- Descriptions to equations 5,6,7 and 8 are added.